# Mathematical Model of Intrinsic Drug Resistance in Lung Cancer

**DOI:** 10.3390/ijms242115801

**Published:** 2023-10-31

**Authors:** Emilia Kozłowska, Andrzej Swierniak

**Affiliations:** Department of Systems Biology and Engineering, Silesian University of Technology, 44100 Gliwie, Poland; emilia.kozlowska@polsl.pl

**Keywords:** lung cancer, mathematical modeling, branching process model, targeted treatment

## Abstract

Drug resistance is a bottleneck in cancer treatment. Commonly, a molecular treatment for cancer leads to the emergence of drug resistance in the long term. Thus, some drugs, despite their initial excellent response, are withdrawn from the market. Lung cancer is one of the most mutated cancers, leading to dozens of targeted therapeutics available against it. Here, we have developed a mechanistic mathematical model describing sensitization to nine groups of targeted therapeutics and the emergence of intrinsic drug resistance. As we focus only on intrinsic drug resistance, we perform the computer simulations of the model only until clinical diagnosis. We have utilized, for model calibration, the whole-exome sequencing data combined with clinical information from over 1000 non-small-cell lung cancer patients. Next, the model has been applied to find an answer to the following questions: When does intrinsic drug resistance emerge? And how long does it take for early-stage lung cancer to grow to an advanced stage? The results show that drug resistance is inevitable at diagnosis but not always detectable and that the time interval between early and advanced-stage tumors depends on the selection advantage of cancer cells.

## 1. Introduction

In the paper, we have developed a multiple-type branching process mathematical model to investigate the emergence of intrinsic drug resistance to the most frequently applied targeted therapeutic drugs in non-small-cell lung cancer. Our focus is on drug resistance in treatment-naive patients. Thus, we performed the simulation only until diagnosis. The model considers nine genes that, if mutated, could sensitize lung cancer patients to a specific group of targeted therapeutics drugs. We have aimed at the quantification of cells that are drug-resistant at diagnosis. The developed model is the most extensive model of targeted treatment in lung cancer as it gathers information about drug sensitivity and drug resistance to nine types of inhibitors. Also, previously researchers have focused only on the process of the emergence of drug resistance and assumed that mutation leading to drug sensitivity is already present in cancer initiating cell.

The developed model that is discrete time has limitations. For example, the model assumes that all cancer cells are synchronized. In addition, the model does not consider interactions between cancer cells. Despite those limitations, the model is accurately calibrated to whole-exome sequencing data. All model limitations are discussed in details in the Discussion section.

Lung cancer is the one with a high mutational burden. Indeed, the number of somatic mutations per megabit in lung cancer is about ten [1,2]. It gives the potential for finding actionable mutations specific to lung cancer cells and could be easily targeted using for example small molecule inhibitors. One of the most frequently targeted genes in lung cancer is *EGFR*, which is mutated in approximately 10% of lung cancer patients [3].

Targeted therapy has revolutionized the treatment of lung cancer [4]. The fraction of patients who receive targeted drugs instead of the standard combination of chemotherapy and radiotherapy (chemoradiation) is expanding drastically. Currently, for about half of non-small-cell lung cancer (NSCLC) patients, targeted therapeutic drugs are available [5].

The bottleneck in the administration of targeted therapeutic drugs is the emergence of drug resistance [6]. Indeed, all targeted therapeutic drugs could cause various types of drug resistance that eventually lead to multi-drug resistance. There are two main types of drug resistance: intrinsic and acquired [7]. The first type is observed as a result of mutation accumulation in the absence of treatment and is subject to investigation in the project. The acquired drug resistance emerges during treatment as a way to protect the tumor from eradication.

Drug resistance in cancer treatment has been present in mathematical biology for about fifty years and started from a seminal work by J. Goldie and A. Coldman [8]. After the discovery of targeted therapeutic drugs, it became a standard problem that mathematical oncology try to tackle. Various mathematical models describing drug resistance to cancer have been developed at various scales: from a molecular model to a whole-tissue one. The most extensive review of standard approaches and methods for modeling drug resistance to cancer was performed by X. Sun and B. Hu [9].

We have investigated, using the model, when the patients develop intrinsic drug resistance—before or after the cancer diagnosis. Using the whole-exome sequencing data from about 1000 non-small-cell lung cancers, we calibrated the developed model. Next, we applied the model to estimate the number of drug-resistant cells at the diagnosis. Lastly, we used the model to estimate the time it takes for an early-stage tumor to grow to an advanced stage.

We estimated that a small fraction of drug-resistant cells are already present at diagnosis. Only when a tumor burden is well before the detection threshold (>1 cm${}^{3}$) is the tumor fully resistant to chemotherapy and targeted therapeutic drugs. However, the fraction of those drug-resistant cells is at most 1% of the whole tumor volume, leading to difficult detection of these mutations, leading to drug resistance.

We also estimated the time interval between early and an advanced tumors. We discovered that in fast-growing tumors, the time is approximately equal to three months in the absence of treatment. This leads to an estimation of maximal time intervals between two consecutive follow-ups.

Our results have important implications in lung cancer treatment. Firstly, all lung cancer patients should be treated with the assumption that they are already drug-resistant. Secondly, lung cancer research should be directed into the eradication of drug-resistant cells rather than their prevention.

## 2. Results

### 2.1. Model Calibration

In the first step, we have applied data from J.D. Campbell et al. [10] to calibrate the developed model. The goal is to fit parameters responsible for the emergence of mutations leading to drug sensitization. The goal is to obtain in computer simulation the same fraction of patients with mutation in the given gene that leads to drug sensitivity as observed in the data.

We have fitted the mathematical model to whole-exome sequencing data. Firstly, we have extracted the percentage of patients with mutated genes from our list. Those data are from treatment-naive patients at the time of diagnosis. Thus, the mathematical model has been simulated only until the diagnosis. A cohort of 10,000 virtual patients has been simulated. From each virtual patient, the percentage of cells with a mutation leading to drug sensitivity in the genes has been computed. Lastly, the mutational signature has been estimated as follows. If the percentage of cells with a mutation leading to drug sensitivity in the gene is equal to or above 5%, then the patient has the gene mutated. As a result, for each patient, we have a binary string, where each digit is one gene. A one in the string means that the gene has mutation leading to drug sensitization and zero if it is not mutated.

The heatmap in Figure 1 shows the mutation signature of 1000 virtual patients at the time of diagnosis. The barplot on the top of the heatmap shows the percentage of patients with a given mutated gene from the calibrated model and the data from J.D. Campbell et al. [10]. For all genes presented on the heatmap, there is no significant difference in percentages between the model and the data. The barplot on the right of the heatmap shows how many genes are mutated in a single patient. As we can see, only a few genes are mutated in a single patient, as also seen in the data.

All the model parameters are listed in Table 1. We assumed that the tumor grows fast so the selection advantage is set to 5%, which is a high value. Mutation rates that lead to drug resistance are set as the value of the probability of point mutation that is equal to $10^{-8}$. Thus, we assume that a single point mutation could lead to drug resistance.

### 2.2. Emergence of Intrinsic Drug Resistance

Next, we applied the calibrated model to investigate the emergence of drug resistance to targeted drugs. The goal is to search for a maximal tumor burden when the tumor is still drug sensitive. This will unravel if it is possible to detect drug resistance already at the diagnosis.

Two model parameters affect the amount of drug-resistant cancer cells in a tumor: the tumor burden at the diagnosis and the probability of gaining one drug-resistance mechanism during a cell division. Thus, we have simulated the model for a wide range of values of these two parameters.

Figure 2 shows the log-normalized number of drug-resistant cells in the tumor as a function of tumor burden at the diagnosis (*M*) and the probability of gaining one drug resistance mechanism during cell division ($u_{resistance}$). As we can see, the number of drug-resistant cells is indeed dependent on the two parameters. As we expected, only a small fraction of cancer cells in the tumor could hold drug resistance. Indeed, when $u_{resistance}=10^{-5}$ and is several folds higher than a probability of point mutation, below 1 cm${}^{3}$ of a tumor has drug resistance when total tumor burden equals 100 cm${}^{3}$.

From the heatmap, we can also notice that drug resistance is inevitable even in a very small tumor that is below the detection threshold (<1 cm${}^{3}$). As we can see, drug resistance in lung cancer emerges when the tumor burden is two/three-fold smaller than the tumor burden at the diagnosis.

### 2.3. Time Interval between Early and Advanced Stage Tumors

The transition between early-stage and late-stage tumors is a turning point in the cancer treatment. The possible treatment interventions are reduced when the tumor stage is high. Thus, it is important to detect cancer when is at an early stage. In the next step, we have computed the time elapsed between early and the late stage lung cancer. The goal is to estimate the minimal time between two consecutive patient follow-ups.

We have performed the following analysis. Firstly, we have simulated the tumor until the tumor size is 1 cm${}^{3}$ (when the tumor is at the early stage). Next, we continue the simulation until the tumor burden is 100 cm${}^{3}$ and is considered a late-stage tumor. Lastly, we have computed the time interval between early-stage and late-stage tumors.

Figure 3 shows the elapsed time between early and late-stage tumors for a wide range of selective advantages. The time interval between early and late-stage tumors varies a lot between low-growing and fast-growing tumor. When selective advantage equals 0.5%, the elapsed time is above two years. However, when selection advantage equals 5%, a tumor grows into to a late-stage tumor within several months.

The selection advantage equals 5% is very high. This speed of tumor growth is observed in the most aggressive tumors. This value is five times higher than the estimated selection advantage in colorectal cancer, which is known for low tumor growth.

Based on the simulation results presented in Figure 3, we can see a time when lung cancer is detectable until it progresses to an advanced stage equals several months. When a patient has aggressive lung cancer, within 3–4 months a tumor could grow to the late stage. Thus, optimally, patients should be screened for lung cancer once every couple of months.

Selection advantage, in contrast to cell division and cell death rate, could be relatively easily calculated using, for example, results from computer tomography (CT). When tumor volume is estimated using CT from two different time points, the selection advantage can be calculated by the percent of the tumor volume increase over a unit of time. However, division rate and death rate could be estimated only using in vitro experiments.

## 3. Discussion

Lung cancer is one of the most mutated types of cancer. Thus, it is an excellent cancer type for searching for novel targeted therapeutic drugs that specifically kill cancer cells. The greatest problem with targeted treatment is that cancer cells develop defense mechanisms against those drugs, leading to drug resistance. Thus, cancer researchers are trying to find a way to overcome drug resistance to different targeted therapeutic drugs.

Mathematical modeling gives the methodology for studying how drug resistance evolves. Dynamical systems could serve as models for the investigation of various methods for overcoming drug resistance. In addition, mathematical modeling could help to predict when drug resistance will emerge. Here, we have aimed to investigate when intrinsic drug resistance will emerge and estimate the optimal time for lung cancer screening.

We predict that intrinsic drug resistance will develop before the tumor becomes detectable. As a result, drug resistance will develop before the tumor burden is above 1 cm${}^{3}$, which is a detection threshold for cancer detection using radiological imaging. The fact that drug resistance is inevitable has implications for cancer treatment. Firstly, it means that the patient should receive, in addition to the main drug, a drug that could resensitize drug-resistant cells. Secondly, all patients should be treated with the assumption that they are already drug-resistant. Thus, the clinical research should be directed into searching for a method to overcome drug resistance instead of establishing drug resistance biomarkers.

The emergence of drug resistance before diagnosis has important implications in the clinic. Firstly, it means that the patient should receive in addition to the main treatment, a drug that could resensitize drug-resistant cells. Secondly, all patients should be treated with the assumption that they are already drug-resistant. Thus, clinical research should be directed into searching for a method to overcome drug resistance instead of establishing drug resistance biomarkers.

We have also predicted that a fraction of drug-resistant cells is several-fold smaller than a tumor burden, leading to the observation that at the diagnosis it is extremely difficult to predict how resistant to a targeted therapeutic drug the tumor is.

We also investigated how long the time interval is between a tumor at an early stage (when the tumor volume equals 1 cm${}^{3}$) and a tumor that is at an advanced stage (when the tumor burden equals 100 cm${}^{3}$). We predicted that a fast-growing tumor could become advanced in about three months. This leads to the conclusion that it is extremely important to start treatment as soon as the tumor is detected.

The major limitations of our results are that we only investigated the tumor dynamics until diagnosis and we did not include the treatment phase in our simulations. From a cohort of patients included for the model calibration, the data about applied treatment are not available. Yet another limitation is the lack of interaction between cancer cells. This was not included in the model because of a lack of appropriate data that could be applied for model calibration. Secondly, the model includes about 10–20 cancer subclones leading to at least 45 additional parameters (in the case of 10 subclones). This leads to a very complex model that is hard to investigate. Lastly, the model assumes that the tumor is a well-mixed system. The development of a spatial model and its calibration to patient data is impossible as the tumor inside the human body occupies even 100 cm${}^{3}$ of space ($10^{12}$ cells). We are far from the possibility of tracking a single cancer cell inside the human body, which is necessary for collecting data for model calibration.

As a future work, we plan the inclusion of a treatment intervention where for each targeted drug data from a separate cohorts of patients should be collected and processed. In addition, we plan to investigate the effect of spatial structure on drug resistance spread.

## 4. Materials and Methods

### 4.1. Pan Lung Cancer Data from J.D. Campbell et al. [10]

We have applied the data from a cohort of non-small-cell lung cancer that includes both squamous and adenocarcinoma cases [10]. The data comprise whole-exome sequencing (WES) of 660 adenocarcinoma/normal pairs and 484 squamous/normal pairs. The cohort of patients includes both previously unpublished data, and the cohort from the TCGA project and Imielinski et al. (2012) [11].

This large cohort allows for a more accurate prediction of the percentage of patients with mutations leading to drug sensitivity in the given gene. What is important, the number of patients with squamous lung cancer and adenocarcinoma is similar. Thus, the cohort is balanced. Next, we filter out patients for which the overall survival and tumor stage is missing, leading to a cohort of 950 patients.

The data were downloaded from CBioPortal on 1 July 2023 [12,13]. All the data were already in a pre-processed form. The downstream analyses were performed in the R environment.

Table 2 shows basic patient characteristics. Most of the patients are males. The most common tumor stage is I and T2, N0, and M0 in the TNM classification. Approximately half of the patients have squamous and the second half the adenocarcinoma subtype of non-small-cell lung cancer.

Using a cohort of 950 patients, we have performed a survival analysis to investigate the survival probability over time, which is important for clinicians. The Kaplan–Meier plot is shown in Figure 4. We have stratified the patients according to the most important predictor of overall survival—tumor stage at the diagnosis. Indeed, for stage I patients, the median overall survival is about 60 months, whereas for stage IV it is only 25 months. As we can see in Figure 4, the *p*-value from the log-rank test is below 0.0001, which means that survival differs significantly between all four groups. We applied the tumor stage from the patient cohort to estimate tumor burden at the diagnosis.

From the cohort, the presence of a mutation in the nine genes was checked. It is assumed that the first somatic mutation in the gene leads to drug sensitivity. Thus, it is enough if one mutation is present. Next, the percentage of patients with at least one mutation in the given gene were calculated and applied for model calibration.

### 4.2. The Branching Processes Mathematical Model

The branching process model is a stochastic process model that could be used to model the process of growth and reproduction [14,15]. It describes how a population of some species, such as cells, bacteria, or viruses, expands and possibly gives rise to a new phenotype [16,17,18]. The most important assumption of this type of model is the lack of interactions between species in a population. One result of lack of interactions between cells is the same rate of reproduction during the whole simulation irrespective of the population size.

The branching processes model is often applied in cancer research and stems from a Goldie–Coldman model that describes the emergence of drug resistance [8,19,20]. It was previously applied to describe tumor evolution [21], the emergence of metastasis [22], and the emergence of drug resistance [23,24,25,26], among others. Even though the model does not take into account interactions between cancer cells and the cancer microenvironment, the model can faithfully describe the emergence of drug resistance. We do not consider cell interactions as we aim at virtual patient simulations and it is impossible to obtain information about cell interactions inside a living body.

Here, we applied the model to describe the process of drug resistance emergence to common molecular treatment drugs. The model was created for non-small-cell lung cancer, which is known for many actionable mutations for which molecular treatment is available.

### 4.3. Virtual Patients Generator

Before the start of a computer simulation, a virtual patient is generated as follows. The tumor stage is sampled from a multinomial distribution where the probability of sampling a specific tumor stage is taken from T in the TNM classification. In Table 2, we have a number of patients for each T tumor stage that reflects the primary tumor size.

The sampled tumor stage is next converted to the tumor burden using a uniform probability distribution. Using the TNM classification, the diameter of a primary tumor is sampled from uniform probability distribution between two values. Table 3 shows the range of the tumor diameter for each T value. Next, assuming a tumor has a spherical shape, tumor volume is computed. Lastly, using a rule that 1 cm${}^{3}$ tumor volume is about $10^{9}$ cells, a tumor burden is computed.

The computation of parameter *M* (tumor burden at the diagnosis) in the above way allows for the generation of a cohort of virtual patients that differ in tumor burden at the diagnosis. As a result, each simulated patient is unique.

### 4.4. The Computer Simulator

We have simulated the model following J. Reiter et al. [27]. The model is a discrete-state and discrete-time model where the system is updated at a constant time interval. Here, the system state is updated once a day. Thus, the simulator assumes that all cells in the system are synchronized. In reality, cancer cells are not synchronized and each cell divides with a different division rate. However, the assumption that the cells are synchronized enables the efficient simulation of large tumors, which is not possible with the classical stochastic simulation algorithm.

The model is implemented in the Julia environment [28]. As the model is not spacial (we assume a well-mixed system) and we update the system at every fixed time interval, the model implementation is efficient computationally.

In the model, tumor cells are growing exponentially with division probability:(1)$$b=\frac{1}{2}\cdot \left(1+s\right),$$
and death probability:(2)d=1−b,
where *s* is the selection advantage of cancer cells. For healthy cells, *s* equals zero, and as a result, the values of death and birth probability are the same. When s>0, the cell population expands, resulting in a tumor. If s=5%, for example, the division probability is bigger than the death probability by 2.5%. So, Equation (Equation 1) incorporates the fact that there are two copies of the same gene.

We do not consider the quiescence of the cells as we are interested in the cell population and not individual cells. Secondly, the inclusion of the quiescence state would only slightly change the quantitative results and would keep the qualitative results the same. This in turn justifies considering only cancer cell division and death. However, to include the quiescence of the cells in the model we may simply decrease parameter *b*.

In addition, division probability is divided into the probabilities of asynchronous divisions (divisions that lead to the emergence of a cell with additional mutation) and synchronous divisions (divisions leading to two cells of the same type). The probability of synchronous division equals:(3)$$b_{synchronous}=b\left(1-u\right)$$
and the probability of asynchronous division equals:(4)$$b_{asynchronous}=b\cdot u$$
where: *u* is the total transition rate from a given subtype. *u* is the sum of all transition probabilities.

At the time (t+1), firstly, the number of cells with a given subclone that die or divide is sampled from a multinomial distribution. In the second step, for each subclone, the number of cells with a given type of division (asynchronous or synchronous) is sampled from multinomial distribution too. Lastly, according to the sampling, the number of cells with a given subclone is updated synchronously.

This simulator allows for an efficient simulation of large tumors but not without drawbacks. The most important limitation is that the simulator assumes that all the cells are synchronized, and thus all cells decide if they divide or die at the same time.

### 4.5. The Mathematical Model of Intrinsic Resistance

We have developed a multiple-type branching process model of lung cancer heterogeneity. The goal of the model is to describe sensitivity and resistance to the most common targeted therapies in lung cancer.

Here, we focus on those targeted drugs that are already approved by the FDA or are in the late clinical trial phase. From Table 2 in Majeed et al., 2021 [29], we extracted a list of nine genes that are mutated in lung cancer patients. The list includes *kras*, *egfr*, *alk*, *met*, *braf*, *ret*, *ros1*, *ntrk*, and *her2*. So, in total, we have included the model nine genes. In addition, the model includes the resistance to platinum-based chemotherapy that is still the standard in the treatment of advanced non-small-cell lung cancer.

In the model, we assume that in a single cancer cell, only one of nine genes, which sensitize cells to targeted treatment, can be mutated. The rationale for this is the fact that it is rare in cancer cells to carry more of these mutations as it leads to cell death. Thus, if one mutation leading to drug sensitivity appears in the cancer cell in the model, all other mutations are blocked. It is known, for example, for EGFR and *KRAS* that their mutations are mutually exclusive [30]. Targeted treatment is usually administered in a monotherapy, and clinicians choose to target this mutation since, according to clinical trials, targeting it leads to the best chance of survival. Thus, clinicians consider only mutations in one gene in treatment planning. There are no available recommendations for clinicians in case the cancer is sensitive to several targeted therapeutic drugs. Therefore, we decided to block the appearance of mutations in other genes when one gene is mutated using the XOR gate.

In addition to a mutation leading to drug sensitization, a cancer cell can also gain resistance to treatment. We have included the process of developing drug resistance as a stochastic process where at each cell division a cancer cell could gain resistance to a drug with a given probability. We assume that even without treatment, the cancer cell could develop resistance. Thus, we consider only intrinsic resistance.

Figure 5 shows the mathematical model schematically. Each cancer cell at each time step performs one of three actions: symmetric division, asymmetric division, or cell death. Symmetric division leads to the appearance of a second cancer cell of the same type. Asymmetric division leads to the emergence of mutation, leading to a new type of cancer cell: subclone. Cancer cell death, however, leads to the removal of one cancer cell from the system.

Lung cancer is initiated by a cell sensitive to platinum-based chemotherapy and resistant to all targeted treatments. During tumor growth, cancer cells firstly gain sensitivity to targeted treatment and then drug resistance. During one cancer cell division, the number of resistance mechanisms can increase only by one. The upper bound for the level of drug resistance does not exist. The maximal level of drug resistance is limited only by the tumor size. This is because more and more mutations leading to drug resistance are accumulating in cancer cells, which in turn leads to multi-drug resistance. Drug resistance is not binary but rather has many levels. It is known that patients can have complete response, partial response, stable disease, or progressive disease. Thus, some patients could have a tumor that is drug-resistant to a given drug only partially.

The model is stochastic, which leads to different mutational signatures for each patient. In addition, the time taken for the emergence of mutation that leads to drug sensitivity or drug resistance is different too.

## Figures and Tables

**Figure 1 ijms-24-15801-f001:**
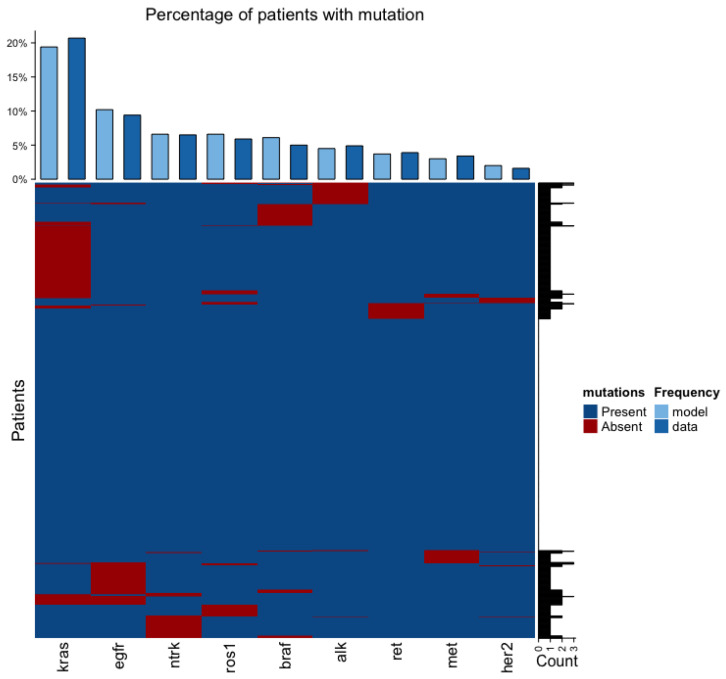
The heatmap shows the cohort of virtual patients, where each row is a single patient. If 5% or more cancer cells have a mutation leading to drug sensitivity in a given gene, then the patient has the mutation in that gene. The barplot on the top of the heatmap shows the fitness of the model compared to the genomics data from lung cancer patients.

**Figure 2 ijms-24-15801-f002:**
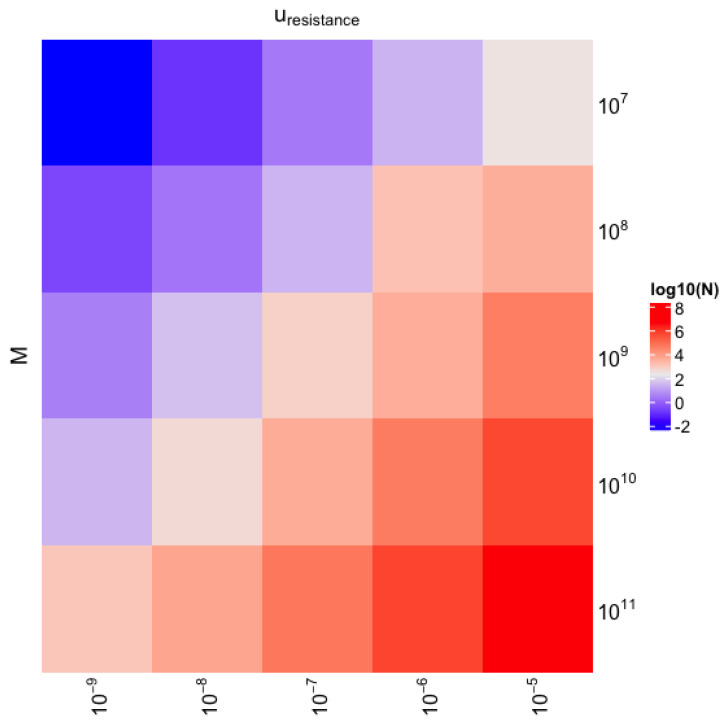
The heatmap presents the log-scaled number of drug-resistant cancer cells at the diagnosis for various values of tumor burden at diagnosis (*M*) and the probability of gaining additional drug resistance mechanisms per cell division $u_{resistance}$.

**Figure 3 ijms-24-15801-f003:**
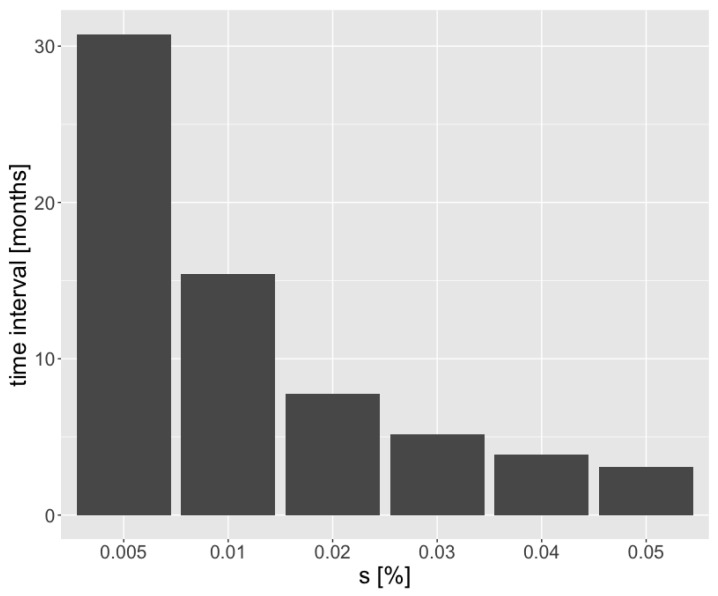
The elapsed time between an early (1 cm${}^{3}$) and a late tumor (100 cm${}^{3}$) as a function of selection advantage. The presented values are medians of 10,000 simulations.

**Figure 4 ijms-24-15801-f004:**
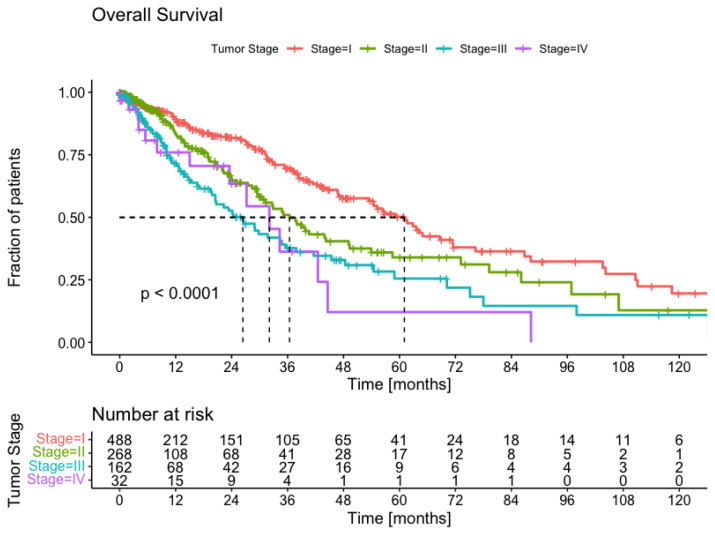
Kaplan–Meier survival plots for lung cancer in the cohort with overall survival as an endpoint. The patients are grouped according to tumor stages, which is a critical clinical parameter defining the patient treatment protocol.

**Figure 5 ijms-24-15801-f005:**
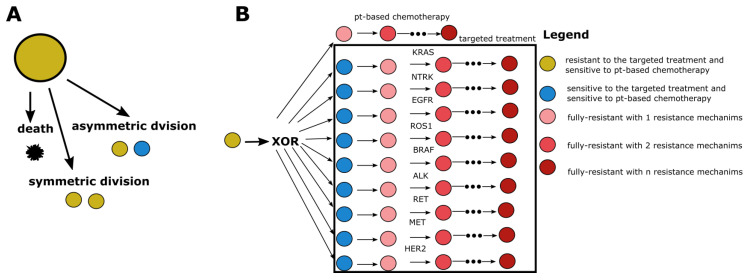
A mathematical model developed herein. (**A**) Each cancer cell, at each time step, can undergo one of three processes: symmetric division, asymmetric division, or death. (**B**) The schema shows the possible transitions between cancer subclones. It is assumed that a single cancer cell can become sensitized to drugs specific to only one gene. The only difference is for platinum-based chemotherapy, to which a tumor-initiating cell is already sensitive. Chemotherapeutic drugs are systemic and attack fast-dividing living cells such as cancer cells. However, a specific driver mutation needs to be accumulated to make the cancer cell sensitive to specific targeted therapeutic drug. The legend shows the meaning of color for each type of circle. pt-based chemotherapy—platinum-based chemotherapy.

**Table 1 ijms-24-15801-t001:** The mathematical model parameters.

Symbol	Value	Name	Reference
*s*	0.05 [%]	selection advantage	assumption
$M_{diagnosis}$	vary with mean equal to $4.26\times 10^{10}$ [cells]	tumor burden at diagnosis	from clinical data
$\mu_{kras}$	$3.15\times 10^{-5}$	probability of mutation leading to drug sensitization in *KRAS*	calibrated
$\mu_{egfr}$	$2.43\times 10^{-5}$	probability of mutation leading to drug sensitization in *EGFR*	calibrated
$\mu_{alk}$	$1.08\times 10^{-5}$	probability of mutation leading to drug sensitization in *ALK*	calibrated
$\mu_{met}$	$1.35\times 10^{-5}$	probability of mutation leading to drug sensitization in *MET*	calibrated
$\mu_{braf}$	$1.80\times 10^{-5}$	probability of mutation leading to drug sensitization in *BRAF*	calibrated
$\mu_{ret}$	$1.35\times 10^{-5}$	probability of mutation leading to drug sensitization in *RET*	calibrated
$\mu_{ros1}$	$1.8\times 10^{-5}$	probability of mutation leading to drug sensitization in *ROS1*	calibrated
$\mu_{ntrk}$	$1.89\times 10^{-5}$	probability of mutation leading to drug sensitization in *NTRK*	calibrated
$\mu_{her2}$	$0.9\times 10^{-5}$	probability of mutation leading to drug sensitization in *HER2*	calibrated
$\mu_{res_{kras}}$	$10^{-8}$	probability of drug resistance to KRASi	assumption
$\mu_{res_{egfr}}$	$10^{-8}$	probability of drug resistance to EGFRi	assumption
$\mu_{res_{alk}}$	$10^{-8}$	probability of drug resistance to ALKi	assumption
$\mu_{res_{met}}$	$10^{-8}$	probability of drug resistance to METi	assumption
$\mu_{res_{braf}}$	$10^{-8}$	probability of drug resistance to BRAFi	assumption
$\mu_{res_{ret}}$	$10^{-8}$	probability of drug resistance to RETi	assumption
$\mu_{res_{ros1}}$	$10^{-8}$	probability of drug resistance to ROSi	assumption
$\mu_{res_{ntrk}}$	$10^{-8}$	probability of drug resistance to NTRKi	assumption
$\mu_{res_{her2}}$	$10^{-8}$	probability of drug resistance to HER2i	assumption
$\mu_{res_{pt}}$	$10^{-8}$	probability of drug resistance to platinum-based chemotherapy	assumption

**Table 2 ijms-24-15801-t002:** Patient’s clinical characteristics in a cohort from J.D. Campbell et al. (2016) [10].

	n = 950	
Sex	Male	567
	Female	383
T Stage	T1	272
	T2	524
	T3	109
	T4	43
	Tx	2
N Stage	N0	603
	N1	215
	N2	109
	N3	7
	Nx	16
M Stage	M0	703
	M1	31
	Mx	209
Stage	I	488
	II	268
	III	162
	IV	32
Subtype	LUAD	479
	LUSC	471
Age *	67 (38–90)	

* the value is shown in the median and in brackets minimum and maximum.

**Table 3 ijms-24-15801-t003:** Translation of T stage into the tumor diameter.

T	Range of Diameter
T1	(0.5 [cm], 3 [cm])
T2	(3 [cm], 5 [cm])
T3	(5 [cm], 7 [cm])
T4	(7 [cm], 10 [cm])

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
