# Peer review of "Mathematical Model of Intrinsic Drug Resistance in Lung Cancer"

_ijms, 2023, doi:10.3390/ijms242115801_

Round 1

Reviewer 1 Report (Previous Reviewer 2)

Comments and Suggestions for Authors

The authors present a revised version of their widthdrawn manuscript on mathematical modelling of resistance in lung cancer. Major criticisms still have not been met.

Since all references are just question marks instead of numbers, a proper review of their manuscript is severely hampered. Before resubmitting, the aujthors should at least fix their references.

The authors must make more visible that their model just covers the time before diagnosis without treatment. Although it is mentioned in the abstract, the current title suggests the modelling of resistance under treatment.

It should be made more clear that the authors are talking about *two* successive mutations: one that makes the cell susceptible and another one that makes it resistant again. "mutated" in l. 25 and 47 means mutated to susceptibility, "mutation" in l. 33 or "mutated" in l. 78 means mutation to resistance. What does "mutated" in l. 1145 or "mutations" in Fig. 2 mean?

Further on, it must be made more clear that only some tumors are sensitive to chemotherapeutics and the "default state" is resistance, like in normal cells. While the introduction (l. 24-25) states that susceptibility is only the exception, the rest of the manuscript suggests that the typical cell is susceptible to all chemotherapeutics.

l. 61-63: A highly simplified mathematical model with mere guesses of the parameters might be interesting for mathematical reasons, it is for sure not accurate enough to derive any medical treatment advice or a serious estimate for the interval between early and advanced tumor.

l. 77-80: If multiple mutations are lethal, the appearance of a second sensitizing mutation (the model seems to start with sensible cells) should not be "blocked" but lead top cell deathg. What does "bocked" mean here, mathematically?

Fig. 1 still shows "wrinkled circles in the legend. These wrinkled circles do not appear anywhere in the figure, there is just a black wrinkled circle which does not appear in the legend.

According to Fig. 2C, the first "mutation" makes, in the cast of pt, resistance (upper branch), but in the case of other therapeutics, sensitivity. This is extremely misleading. >So, the "XOR-gate" means that either the cell gains pt resistance, or sensitivity to one of the chemotherapeutics?

The emergence of a sensitizing mutation in Fig. 1B would not make the complete tumor (or "the patient" sensitive, but just some cells. The rest of the tumor would stay resistant. Mutations are an event on cellular, not on patient basis, and different cells in obne patient are sensitive or resistant to different chemotherapeutics.

What does "mutated" in Fig. 2 mean? There should be three states: "default" state resistant to chamotherapeutics, mutated to sensitivity and mutated further on back to resistance. How did you determine from the Campbell data which mutation leads to sensitivity and which leads to resistance?

l. 134 "emergence of resistance" you should determine the emergence of *sensitivity* first.

l. 301 To -> to

l. 32f: two types of drug resistance COMMA intrinsic and aquired

Author Response

    1. Since all references are just question marks instead of numbers, a proper review of their manuscript is severely hampered. Before resubmitting, the aujthors should at least fix their references.

    We have double-checked if TeX file is properly compiled with a bibliography. Frankly speaking, we do not know how it has happened that the reviewer has obtained the previous version with question marks instead numbers.

    2. The authors must make more visible that their model just covers the time before diagnosis without treatment. Although it is mentioned in the abstract, the current title suggests the modelling of resistance under treatment.
    We slightly modified the title to: “Mathematical Model Of Intrinsic Drug Resistance In Lung Cancer” to make it more clear that we investigated drug resistance in treatment-naïve patients. We have also clarify this issue in the Introduction and the Results sections.

    1. It should be made more clear that the authors are talking about *two* successive mutations: one that makes the cell susceptible and another one that makes it resistant again. "mutated" in l. 25 and 47 means mutated to susceptibility, "mutation" in l. 33 or "mutated" in l. 78 means mutation to resistance. What does "mutated" in l. 1145 or "mutations" in Fig. 2 mean?
      We have introduced the following terminology in the paper: mutation leading to sensitivity and mutation leading to drug resistance. We differentiate those two types of mutations and we have consequently used this terminology.

    2. Further on, it must be made more clear that only some tumors are sensitive to chemotherapeutics and the "default state" is resistance, like in normal cells. While the introduction (l. 24-25) states that susceptibility is only the exception, the rest of the manuscript suggests that the typical cell is susceptible to all chemotherapeutics.
      We agree that there are cancers that are not sensitive to chemotherapy. However, non-small cell lung cancer is considered to be usually sensitive to platinum-based chemotherapy. On the other hand, lung cancer is resistant to targeted therapeutic drugs unless a mutation in the targeted gene is present. We have now clarified this in the paper.

    3. l. 61-63: A highly simplified mathematical model with mere guesses of the parameters might be interesting for mathematical reasons, it is for sure not accurate enough to derive any medical treatment advice or a serious estimate for the interval between early and advanced tumor.

    We agree that our simplified model is not accurate enough to recommend quantitively protocols of drug administration. Nevertheless, it is sufficient to formulate qualitative recommendations and qualitative estimation of the interval between early and advanced tumor. On the other hand, it is possible to measure the selection advantage in patients whereas birth and death rates can be only measured using in vitro methods.

    1. l. 77-80: If multiple mutations are lethal, the appearance of a second sensitizing mutation (the model seems to start with sensible cells) should not be "blocked" but lead top cell deathg. What does "bocked" mean here, mathematically?

    It is known that at least some multiple mutations leading to drug sensitivity cause cell death. We refer in the paper to some reports confirming this statement. Generally, it is not proved that it is always the case, however clinicians have no recommendation what to do with several sensitizing mutations and the first discovered mutation decides about drug administration. Thus, from medical point of view, the computer simulations are performed in such a way that cancer cells with multiple mutations leading to drug sensitivity die. Mathematically, this phenomenon is included in the model by inclusion of XOR logical gate.

    1. Fig. 1 still shows "wrinkled circles in the legend. These wrinkled circles do not appear anywhere in the figure, there is just a black wrinkled circle which does not appear in the legend.

    We have modified Fig. 1 (now, it is Figure 5) in such a way that wrinkled circles in the legend are replaced with circles. It is to show that the circles represent the colors and colors represent different cancer subclones.

    1. According to Fig. 2C, the first "mutation" makes, in the cast of pt, resistance (upper branch), but in the case of other therapeutics, sensitivity. This is extremely misleading. >So, the "XOR-gate" means that either the cell gains pt resistance, or sensitivity to one of the chemotherapeutics?

    Yes, the first cancer cell is sensitive to platinum-based chemotherapy and is resistant to all targeted therapeutic drugs. Now, in Figure 5, we have shown it more clearly to make sure this is not misleading.

    1. The emergence of a sensitizing mutation in Fig. 1B would not make the complete tumor (or "the patient" sensitive, but just some cells. The rest of the tumor would stay resistant. Mutations are an event on cellular, not on patient basis, and different cells in obne patient are sensitive or resistant to different chemotherapeutics.
      We agree. Thus, we removed Figure 2B (now, Figure 5) that has been misleading.

    2. What does "mutated" in Fig. 2 mean? There should be three states: "default" state resistant to chamotherapeutics, mutated to sensitivity and mutated further on back to resistance. How did you determine from the Campbell data which mutation leads to sensitivity and which leads to resistance?
      We assumed that the first mutation in the selected genes leads to drug sensitivity. Thus, it is enough if a patient has one mutation to consider him/her as drug-sensitive to targeted therapeutic drugs and it was determined from Campbell data.

    q
    11. l. 134 "emergence of resistance" you should determine the emergence of *sensitivity* first.
    We determine the emergence of drug sensitivity in Figure 1.

    1. l. 301 To -> to

    Thank you for noticing this typographical error. We have corrected it.

    1. l. 32f: two types of drug resistance COMMA intrinsic and aquired

    Thank you for noticing this punctuation error. We have corrected it.

Reviewer 2 Report (Previous Reviewer 3)

Comments and Suggestions for Authors

After reviewing the paper, here are some feedbacks on three areas that could be improved:

Firstly, the abstract is too lengthy and includes unnecessary details that do not contribute to the main message. A shorter and more concise abstract would enable readers to quickly grasp the main idea and motivation of the paper. For instance, the abstract could be summarized as follows:

This paper presents a mathematical model of drug resistance to targeted treatment in lung cancer. The model is a multiple-type branching process that describes the emergence of intrinsic drug resistance to nine groups of targeted therapeutics. The model is calibrated to whole-exome sequencing data from over 1,000 non-small cell lung cancer patients. The model is used to investigate when intrinsic drug resistance emerges and how long it takes for an early-stage tumor to grow to an advanced stage. The results show that drug resistance is inevitable at diagnosis, but not always detectable, and that the time interval between early and advanced stage tumors depends on the selection advantage of cancer cells.

Secondly, the introduction lacks a clear statement of the research question and the main contribution of the paper. The introduction should provide a brief overview of the background, motivation, objectives, methods, and results of the paper. It should also highlight the novelty and significance of the paper in relation to the existing literature. For example, the introduction could start with something like:

Drug resistance is a major challenge in cancer treatment, especially for lung cancer, which is one of the most mutated cancers. Targeted therapy has revolutionized the treatment of lung cancer by exploiting specific mutations that sensitize cancer cells to certain drugs. However, targeted therapy also leads to the emergence of drug resistance, either intrinsic or acquired, which reduces the efficacy and duration of treatment. Therefore, it is important to understand how and when drug resistance emerges and how it affects tumor growth and progression. In this paper, we develop a mechanistic mathematical model of drug resistance to targeted treatment in lung cancer and use it to address the following questions:

Lastly, the results section is not well-organized and does not present the main findings clearly. The results section should be structured according to the research questions and hypotheses, and provide sufficient evidence and explanation for each result. The results section should also include figures and tables that illustrate the results visually and support the claims made in the text. For example, the results section could be divided into subsections such as:

2.1 Model calibration 

2.2 Emergence of intrinsic drug resistance 

2.3 Time interval between early and advanced stage tumors

Each subsection should start with a brief introduction of what is being investigated, followed by a description of the methods and parameters used, then a presentation and interpretation of the results, and finally a discussion of the implications and limitations of the results.

Comments on the Quality of English Language

The article is written in a clear and coherent manner, utilizing appropriate technical terms and academic style. The article follows the standard structure of an academic paper, including an abstract, introduction, results, discussion, materials and methods, and references sections. The article employs proper grammar, punctuation, and spelling throughout, with only a few minor errors that do not impede the readability or comprehension of the text.

The article employs a variety of sentence structures and transitions to effectively convey the main points and arguments. The article also uses active and passive voice appropriately, depending on the context and the focus of the sentence. The article avoids unnecessary repetition and redundancy, utilizing synonyms and paraphrases to avoid monotony. The article also uses modifiers and qualifiers to express the degree of certainty or uncertainty of the statements.

The article cites relevant sources to support the claims and findings, utilizing a consistent citation style (in this case, MDPI style). The article acknowledges the limitations and implications of the research, and provides suggestions for future work. The article uses figures and tables to visually illustrate the results and support the claims made in the text. The article also provides captions and legends for the figures and tables, and refers to them in the text.

Overall, the article demonstrates a high level of English proficiency, comparable to that of a native speaker or an advanced learner. The article is well-written and well-organized, effectively communicating the research to the intended audience. The article could benefit from some minor proofreading and editing to correct some typos and improve some word choices, but these do not detract from the quality of the paper.

Author Response

  1. Firstly, the abstract is too lengthy and includes unnecessary details that do not contribute to the main message. A shorter and more concise abstract would enable readers to quickly grasp the main idea and motivation of the paper. For instance, the abstract could be summarized as follows:

This paper presents a mathematical model of drug resistance to targeted treatment in lung cancer. The model is a multiple-type branching process that describes the emergence of intrinsic drug resistance to nine groups of targeted therapeutics. The model is calibrated to whole-exome sequencing data from over 1,000 non-small cell lung cancer patients. The model is used to investigate when intrinsic drug resistance emerges and how long it takes for an early-stage tumor to grow to an advanced stage. The results show that drug resistance is inevitable at diagnosis, but not always detectable, and that the time interval between early and advanced stage tumors depends on the selection advantage of cancer cells.

We have shortened the abstract to make it more concise and to follow the reviewer’s suggestion.

  1. Secondly, the introduction lacks a clear statement of the research question and the main contribution of the paper. The introduction should provide a brief overview of the background, motivation, objectives, methods, and results of the paper. It should also highlight the novelty and significance of the paper in relation to the existing literature. For example, the introduction could start with something like:

Drug resistance is a major challenge in cancer treatment, especially for lung cancer, which is one of the most mutated cancers. Targeted therapy has revolutionized the treatment of lung cancer by exploiting specific mutations that sensitize e cancer cells to certain drugs. However, targeted therapy also leads to the emergence of drug resistance, either intrinsic or acquired, which reduces the efficacy and duration of treatment. Therefore, it is important to understand how and when drug resistance emerges and how it affects tumor growth and progression. In this paper, we develop a mechanistic mathematical model of drug resistance to targeted treatment in lung cancer and use it to address the following questions:

We have modified the Introduction section by including an additional paragraph describing the significance and novelty of the paper in relation to existing literature: “Our results have important implications in lung cancer treatment. Firstly, all lung cancer patients should be treated with the assumption that they are already drug-resistant. Secondly, lung cancer research should be directed into the eradication of drug-resistant cells rather than its prevention.” Moreover, we have changed the structure of the Introduction in such a way to clearly state the research question and the main contribution of the paper.

  1. Lastly, the results section is not well-organized and does not present the main findings clearly. The results section should be structured according to the research questions and hypotheses, and provide sufficient evidence and explanation for each result. The results section should also include figures and tables that illustrate the results visually and support the claims made in the text. For example, the results section could be divided into subsections such as:

2.1 Model calibration

2.2 Emergence of intrinsic drug resistance

2.3 Time interval between early and advanced stage tumors

Each subsection should start with a brief introduction of what is being investigated, followed by a description of the methods and parameters used, then a presentation and interpretation of the results, and finally a discussion of the implications and limitations of the results.

We have rearranged the Results section and have modified the subheadings. Now, each subsection after the introduction contains the research question and hypothesis, a description of the methods and parameters used, a presentation and interpretation of the results, and finally a discussion of the implications and limitations of the results. We also moved the model description to the Methods section.

  1. Overall, the article demonstrates a high level of English proficiency, comparable to that of a native speaker or an advanced learner. The article is well-written and well-organized, effectively communicating the research to the intended audience. The article could benefit from some minor proofreading and editing to correct some typos and improve some word choices, but these do not detract from the quality of the paper.

We have read the article many times and we have done our best to correct all grammatical errors and typos.

This manuscript is a resubmission of an earlier submission. The following is a list of the peer review reports and author responses from that submission.

Round 1

Reviewer 1 Report

Comments and Suggestions for Authors

This work proposed a mechanistic mathematical model to investigate the emergence of drug resistance in lung cancer patients. The manuscript provided detailed description of the model development and calibration, supported by whole-exome sequencing data from non-small cell lung cancer patients. The research is well-structured and the methodology is well-explained. The results demonstrated the inevitability of drug resistance at the diagnosis stage, even when the probability of resistance emergence is relatively low. The findings also highlighted the challenge of detecting resistance due to the low fraction of drug-resistant cells.

There are a few areas that could benefit from further clarification and discussion. Firstly, the limitations of the study should be discussed, particularly regarding the assumption of lack of interactions among cells in the branching process model. This assumption might not accurately represent the complex interactions within a tumor microenvironment. Additionally, it would be valuable to address the implications of the finding that drug resistance is inevitable at diagnosis. How might this affect treatment strategies and patient outcomes? Furthermore, the study could be strengthened by considering the impact of various treatment interventions on the emergence and spread of drug resistance. Discussing potential strategies to delay or prevent resistance emergence under different treatment scenarios would enhance the practical implications of the research.

In conclusion, the paper presented a well-executed mathematical model to study drug resistance in lung cancer. The results provided valuable insights into the timing of resistance emergence and its implications for patient management. Addressing the mentioned points would enhance the significance and applicability of the research in the context of clinical decision-making and treatment strategies.

Reviewer 2 Report

Comments and Suggestions for Authors

Kozlowska and Swierniak present a mathematical model to simulate mutations in lung cancer cells. The manuscript is not clearly written and contains serious flaws, the methods used not appropriate and the whole manuscript not suitable for publication.

The authors do not clearly outline their goals. Just in the discussion section (l. 264f) it is made clear that the authors just want to simulate the mutations until diagnosis, i.e. without treatment. That was not made clear before.

Why do the authors present survival plots (Fig. 1) if they are not included in or compared to the model at all? Also the grades of the tumor (Tab. 1) are not used. The only data from the Campbell study used in the model (for calibration) is, for each of the nine genes, the percentage of patients with a mutation.

It is very questionable that in every step, a cell can only either divide or die. It might also just stay alive which is not covered in the model. A guess of one day per replication cycle is for sure not accurate enough to draw any conclusions about the time frame of tumor growth (l. 261f).

That all species reproduce at the same rate is not the "result" of the "lack of interactions" between species (l. 91ff) but rather an independent setting.

Why is the division probability expressed at 1/2*(1+s) (eq 1)? Wat is the meaning of "s", why is the division rate not modeled directly?

eq 4 should read "b_asyncronous".

It seems that the authors assume two kinds of mutation, a "mutation leading to drug sensitization" and a gain of resistance (l. 143f). That would mean at the beginning the cells are resistant, then susceptible and then resistant again? That does not make sense. Plus, the data shown includes only mutated yes vs no. Does "mutated" mean mutated to resistance or to susceptibility?

Fig. 2B shows time lines of "patients". However, a mutation is an event on cell level, not on patient level.

The legend of Fig. 2C shows wrinkled circles but Fig. 2C shows only smooth circles. Why are there three (1, 2, n) different counts of "resistance mechanisms"? Why does Fig. 2C show "pt-basefd chemotherapy" while therapy is not modelled at all? Why is the first circle on the left (KRAS) pink but all others blue?

Why is there no upper bound of resistance (l. 157f)? Resistance should be binary, present or not.

What is the "pt" gene in Fig. 3 on the left? Does "mutation" mean sensitive or resistant? Why is the pt gene missing in the right part of Fig. 3?

Is M (tumor burden at diagnosis) 10^11 (l. 186) or 10^10 (Tab. 2)?

What are "the rest of the parameters" (l. 190)?Why is the probabilitzy of mutation for mu_kras and mu_egfr "assumed" and not calculated as the others (Tab. 2)? Where do these assumptions come from? Where does 10^-8, the "assumption" for all the mu_res, come from? In which equation are they used?

Reviewer 3 Report

Comments and Suggestions for Authors
  • One limitation of the paper is that it assumes that only one mutation in one of the nine genes can sensitize a cancer cell to a specific targeted therapy, and that multiple mutations are lethal to the cell. This assumption may not reflect the true complexity and heterogeneity of lung cancer, as there are cases where cancer cells can harbor more than one mutation in different genes, or even co-occurring mutations in the same gene, that may affect their response to treatment . For example, some studies have reported that EGFR-mutant lung cancer patients may also have mutations in KRAS, MET, HER2, or BRAF, which can confer resistance to EGFR inhibitors or other targeted therapies . Therefore, the paper could have explored more realistic scenarios where cancer cells can have multiple mutations that may interact or interfere with each other, and how this may affect the emergence and dynamics of drug resistance.
  • Another limitation of the paper is that it does not consider the spatial structure and microenvironment of the tumor, which may also influence the evolution and selection of drug-resistant clones. The paper uses a well-mixed branching process model, which assumes that all cells are equally exposed to the drug and have equal chances of dividing or dying. However, in reality, tumors are often composed of different regions or compartments that may have different levels of oxygen, nutrients, immune cells, stromal cells, and drug penetration . These factors may create different selective pressures and niches for drug-resistant cells to emerge and proliferate . Therefore, the paper could have incorporated some spatial aspects or heterogeneity into the model, such as using a cellular automaton or agent-based model, to capture the effects of tumor microenvironment on drug resistance.
  • A third limitation of the paper is that it does not validate or compare its model predictions with experimental or clinical data. The paper only calibrates its model parameters based on the frequency of mutations in nine genes from a cohort of lung cancer patients at diagnosis. However, this does not necessarily reflect how these mutations affect the response and resistance to different targeted therapies in vivo or in vitro. The paper also does not test its model predictions against any independent data sets or outcomes, such as survival rates, progression-free survival, tumor growth rates, or molecular profiles of drug-resistant tumors . Therefore, the paper could have improved its model validation and verification by using more relevant and reliable data sources and metrics to evaluate its model performance and accuracy.
Comments on the Quality of English Language

The paper has a clear and logical structure, with an introduction, materials and methods, results, and discussion sections. The paper also provides a comprehensive list of references and a table of model parameters. The paper uses appropriate scientific terminology and follows the conventions of mathematical notation. The paper also includes some figures and tables to illustrate the data and the model predictions.

However, the paper also has some language issues that affect its readability and clarity. Here are some examples of the language issues and some suggestions for improvement:

  • In the abstract, the sentence “As a result, some drugs, despite of initial excellent response, are withdrawn from the market.” should be “As a result, some drugs, despite their initial excellent response, are withdrawn from the market.” The preposition “of” is not needed after “despite”, and the pronoun “their” is needed to refer to the drugs.
  • In the introduction, the sentence “One of the most targeted genes in lung cancer is EGFRwhich is mutated in approximately 10% of lung cancer patients [3].” should be “One of the most frequently targeted genes in lung cancer is EGFR, which is mutated in approximately 10% of lung cancer patients [3].” The word “frequently” is needed to modify “targeted”, and there should be a comma after “EGFR”.
  • In the materials and methods section, the sentence “The data were downloaded from CBioPortal on 1Pst July 2023 [12,13].” should be “The data were downloaded from CBioPortal on 1st July 2023 [12][13].” There should be no capital P in “1st”, and there should be no comma between the references.
  • In the results section, the sentence “We have predicted that intrinsic drug resistance develops before the tumor is detectable.” should be “We have predicted that intrinsic drug resistance develops before the tumor becomes detectable.” The verb “becomes” is needed to indicate the change of state.
  • In the discussion section, the sentence “Thus, we assume that developing mutations leading to drug resistance to more than one drug is lethal to cancer cells.” should be “Thus, we assume that developing mutations that lead to drug resistance to more than one drug is lethal to cancer cells.” The relative pronoun “that” is needed to introduce the subordinate clause.

These are some examples of language issues that could be improved in the paper. There are also some minor spelling and punctuation errors throughout the paper that could be corrected by using a spell-checker or proofreading tool. Overall, the paper demonstrates a good level of English proficiency, but it could benefit from some editing and revision to enhance its readability and clarity